# Influences of Cu Doping on the Microstructure, Optical and Resistance Switching Properties of Zinc OxideThin Films

**DOI:** 10.3390/nano13192685

**Published:** 2023-09-30

**Authors:** Jun-Hong Weng, Ming-Cheng Kao, Kai-Huang Chen, Men-Zhe Li

**Affiliations:** 1Department of Electrical Engineering, Tunghai University, Taichung 40704, Taiwan; jhw@thu.edu.tw; 2Department of Information and Communication Engineering, Chaoyang University of Technology, Taichung 413310, Taiwan; 3Department of Electronic Engineering, Center for Environmental Toxin and Emerging-Contaminant Research, Super Micro Mass Research & Technology Center, Cheng Shiu University, Chengcing Rd., Niaosong District, Kaohsiung 83347, Taiwan; 5977@gcloud.csu.edu.tw; 4Department of Electronic Engineering, Hsiuping University of Science and Technology, Taichung 41280, Taiwan

**Keywords:** ZnO, sol-gel, thin films, resistive switching

## Abstract

Copper-doped zinc oxide films (Zn_1−x_Cu_x_O) (x = 0, 2%, 4%, 6%) were fabricated on conductive substrates using the sol-gel process. The crystal structure, optical and resistive switching properties of Zn_1−x_Cu_x_O films are studied and discussed. RRAM is made using Zn_1−x_Cu_x_O as the resistive layer. The results show that the (002) peak intensity and grain size of Zn_1−x_Cu_x_Ofilms increase from 0 to 6%. In addition, PL spectroscopy shows that the oxygen vacancy defect density of Zn_1−x_Cu_x_O films also increases with the increase in Cu. The improved resistive switching performance of the RRAM device can be attributed to the formation of conductive filaments and the destruction of more oxygen vacancies in the Zn_1−x_Cu_x_O film. Consequently, the RRAM device exhibits a higher low resistance state to high resistance state ratio and an HRS state of higher resistance value.

## 1. Introduction

In recent research, resistive random access memory (RRAM) devices have been developed using various binary oxide materials, making them highly attractive for next-generation memory applications due to their fast operation, simple structure, low power consumption, nondestructive readout, long data retention, ease of fabrication, and cost-effectiveness [1,2,3,4,5]. Among the different oxide materials, ZnO-based semiconductor thin films are widely employed in RRAM devices. The ZnO-based semiconductor was investigated for its potential applications in RRAM, due to its low carrier trapping degree, non-toxicity, and simple chemical composition [6,7,8,9,10].Recent studies have indicated that doping zinc oxide with transition metals can effectively reduce its intrinsic carrier density [11,12]. Specifically, La doping has been found to decrease the oxygen vacancy density in ZnO thin films, leading to an improved ON/OFF ratio in RRAM devices [13]. Numerous studies have also demonstrated RRAM characteristics in various doped ZnO thin films [14,15,16,17,18]. Additionally, researchers have explored Cu-doped ZnO materials, the grain size of which can be controlled by the amount of copper doping [19,20]. Cu is considered a suitable dopant for ZnO due to its similar ionic radii to Zn and its ability to act as a deep receptor, binding with adjacent O vacancies in the ZnO lattice [21,22].

We investigate the effects of copper on the electrical, structural, and switched resistive properties of Zn_1−x_Cu_x_O thin films. The Zn_1−x_Cu_x_O was synthesized on FTO substrates through the sol-gel process, followed by rapid thermal treatment. The crystallization and microstructure of the samples were analyzed using an X-ray diffractometer and a scanning electron microscope, respectively. RRAM devices employing Zn_1−x_Cu_x_O thin films were thoroughly examined, and their bipolar switching characteristics were analyzed and discussed.

## 2. Experimental Section

In this research, we prepared thin films of Zn_1−x_Cu_x_O (x = 0%, 2%, 4%, 6%) using the sol-gel process.Zn_1−x_Cu_x_O films were prepared using a sol-gel method, employing zinc acetate dihydrate (CH3COO)_2_Zn·2H_2_O and copper acetate monohydrate (Cu(CH_3_COO)_2_·2H_2_O) as precursor materials. The precursors were dissolved in 2-Methoxyethanol solvent to maintain a constant metal ion concentration of 0.05 M. Ethanolamine was added to ensure solution stability. These solutions were stirred at 60 °C for three hours using a magnetic stirrer and then allowed to stand for 72 h to obtain transparent solutions. The sol-gel solution was deposited onto a substrate mounted on a spin coater using a two-step spin process. The deposition step began with spinning at 1000 rpm for 10 s and then at 3000 rpm for 30 s. Subsequently, the samples were annealed on a hot plate at 300 °C for 2 min to remove the solvent. The entire process was repeated ten times to achieve the desired film thickness. Finally, the film was annealed at 600 °C for 2 min in a rapid annealing system. The annealing rate was 100 °C/min. The thickness of Zn_1−x_Cu_x_O film was 500 nm withα-step surface profiler.

Crystallization of samples was analyzed using XRD. To study the optical properties of Zn_1−x_Cu_x_O, we conducted transmittance optical measurements using a UV spectrometer. The effect of the fluorine-doped tin oxide (FTO) substrate was also taken into account during the transmittance correction. This is due to the fact that FTO substrate can absorb light in the UV-visible range. The transmittance may be measured inaccurately. We used a blank FTO substrate (without Zn_1−x_Cu_x_O sample) for reference measurements. To correct for substrate effects, the transmittance of the FTO substrate alone was subtracted from the transmittance of the sample and substrate combination. These corrected transmittance data represent the transmittance of the Zn_1−x_Cu_x_O sample without FTO substrate contribution. For analyzing optical emissions, photoluminescence spectroscopy was employed, covering a wavelength range from 350 nm to 600 nm. In addition, we characterized the switching properties of current-voltage for the Zn_1−x_Cu_x_O thin films in RRAM. RRAM with a Ag/Zn_1−x_Cu_x_O/FTO structure was measured using a semiconductor parameter analyzer. Figure 1 shows a schematic diagram of an RRAM device with Ag electrodes. One hundred circular top silver electrodes (100 nm thickness and 0.05 cm diameter) were fabricated on top of the Zn_1−x_Cu_x_O layer using thermal evaporation technique at 5 × 10^−6^ torr.

## 3. Results and Discussion

Figure 2 displays the XRD for Zn_1−x_Cu_x_O with various Cu concentrations, compared with the pure zinc oxide. The XRD patterns match the standard JCPDS data (36-1451), indicating the absence of a rutile phase or any other phase. All samples have a wurtzite hexagonal structure oriented along the c-axis and a symmetric P63/mc structure. There is no evidence of phase separation in the Zn_1−x_Cu_x_O thin films with the replacement of Zn with Cu ions. In addition, there are no diffraction peaks of other impurities or compounds, indicating that copper ions have been doped into the ZnO lattice to replace zinc ions. The (002) peak intensity of Zn_1−x_Cu_x_O films increases with the Cu doping content from 2% to 6%. The samples exhibit a (002) peak of ZnO at approximately 2θ = 34.5°. The enhanced (002) peak intensity with a higher Cu concentration can be attributed to lattice strain introduced by the size difference between Cu and Zn ions when Cu ions replace Zn ions in the ZnO lattice. The increased intensity of the (002) diffraction peak at approximately 2θ = 34.5° indicates a more ordered and periodic arrangement of atoms in the lattice along the c-axis. Lattice strain caused by the size mismatch between copper and zinc ions leads to this increase in order. When the atomic planes along the c-axis are more regularly spaced, the X-rays diffract more efficiently, resulting in higher intensity peaks in the XRD pattern. Strain affects the crystal structure and orientation of ZnO, resulting in greater distortion and increased (002) peak intensity as the Cu concentration rises. 

FWHM reflects the change in the average size of Zn_1−x_Cu_x_O grains. The grain size of Zn_1−x_Cu_x_Odecreases with increasing FWHM. Specifically, a slight increase in FWHM was observed as the Cu concentration increased from 0% to 6%. Average sizes of grains for films, calculated using the Scherrer formula, were 20.3 nm for 0%, 15.2 nm for 2%, 9.5 nm for 4%, and 8.6 nm for 6%, respectively [23]. Figure 3 displays the surface morphology of theZn_1−x_Cu_x_O films, showing granular porous surfaces for all samples. It is evident that grain size decreases with the increase in the Cu concentration. This phenomenon can be attributed to the crystal structure and growth of the material. When copper is introduced into the zinc oxide lattice, copper (Cu) and zinc (Zn) have different atomic sizes and lattice constants, creating a lattice mismatch. The lattice mismatch in the Zn_1−x_Cu_x_Ofilm causes crystal strain in the crystal structure and hinders the growth of larger grains. In addition, the surface density of films increases as the Cu concentration rises. Furthermore, the thickness of all samples was found to be 500 nm.

Figure 4 shows the photoluminescence (PL) spectrum of Zn_1−x_Cu_x_O at room temperature. The PL spectrum of ZnO exhibits two main components: one in the ultraviolet region and the other in the yellow-green visible region. The ultraviolet emission band at 385 nm shows the transition of electrons from the bottom of the conduction band to the zinc vacancy energy level [24,25]. Multiple yellow–green emissions observed in Zn_1−x_Cu_x_O films are attributed to oxygen vacancy defects, which correspond to the transition from the conduction band to oxygen vacancy defects [26]. The ultraviolet emission at 385 nm of the Zn_1−x_Cu_x_O thin film increases as the copper content increases. This may be due to the copper doping causing more zinc vacancy energy levels in the ZnO thin film, which enhances the ultraviolet emission. In addition, the yellow–green light emission of Zn_1−x_Cu_x_O thin film increases with the increase in the copper doping amount. This enhancement may be caused by the replacement of copper ions into the ZnO lattice to form Zn_1−x_Cu_x_O, and more copper ions generate more oxygen vacancy defects in zinc oxide. Therefore, the transition from the conduction band to oxygen vacancy defects will also increase, resulting in an increase in yellow–green light emission in the Zn_1−x_Cu_x_O thin film. 

Transmission spectra with a range of 300–800 nm for Zn_1−x_Cu_x_O are shown in Figure 5. As the doping of Cu increases from 0% to 6%, the light transmittance for the Zn_1−x_Cu_x_O film decreases from 92% to 78%. To determine the absorption coefficient and bandgap of these films, graphical methods were employed. The value of (αhυ)^2^ can be obtained from the equation αhυ = A(hυ-Eg)n. In addition, the value of α can be obtained using the equation α = –(1/D)lnT [27], where T is the transmittance, A is a function of the refractive index and hole/electron effective masses, h is the constant of Planck, v is the frequency of photons, and Eg is the bandgap. For a direct bandgap semiconductor (*n* = 1/2), the bandgaps of all films were calculated via the extrapolation of the straight portion for the curve of (*αhυ*)^2^ = 0 in Figure 6.The inset of Figure 6 shows the relationship between the Cu doping amount and sample band gap energy. As the Cu content increases from 0 to 6%, the bandgap inZn_1−x_Cu_x_Ofilms decreases from 3.285 eV to 3.271 eV. This phenomenon may be attributed to the change in the energy band structure in the ZnO film when Cu ions are incorporated into the ZnO lattice. An exchange interaction occurs between the valence band/conduction band electrons of ZnO and the local d-electrons of Cu ions replacing Zn ions. This exchange interaction introduces a new shallow energy level between the valence band and conduction band of ZnO, resulting in a reduction in the band gap of ZnO [28]. The transmittance decreases with increased Cu content, and the bandgap energy reduces as Cu content increases. The observed correlation between the bandgap and Cu content is likely influenced by the microstructure changes arising from the incorporation of Cu ions.

Figure 7 illustrates the bipolar I-V switching curves of Zn_1−x_Cu_x_O films during the initial formation at 5 V in RRAM, with green arrows indicating the direction of the voltage sweep. The compliance current is controlled at 10 mA. RRAM devices can be operated using LRS-HRS switching cycles up to 100 times without losing the electric properties. The real image of Zn_1−x_Cu_x_O films’ RRAM device is shown in Figure 7a. The RRAM setup process involves shifting the device into the LRS with a high forward bias of the control voltage. The reset process is to reduce the operating current of the LRS and provide a negative bias voltage to the reset voltage to transition to the HRS state, showing a bipolar resistance change characteristic. According to the bipolar I-V switching curve of the Zn_1−x_Cu_x_O film in Figure 7, the set voltages are 1.1V (0%), 1.2V (2%), 3V (4%), and 1.5V (6%), respectively. In addition, the rest voltages are −0.9V (0%), −1.4V (2%), −2.8V (4%), and −1.5V (6%), respectively. The LRS and HRS properties of RRAM can be explained by the formation and destruction of conductive filaments composed of oxygen vacancies [2,3,4,5]. The switching of RRAM can be explained by two resistance states (HRS and LRS), the basic mechanism of which is shown in Figure 8. The SET process is initiated when an external positive voltage is applied to the top electrode (Ag), as shown in Figure 8. During this process, oxygen anions (O^2−^) migrate away from their original sites towards the top electrode, creating oxygen vacancies in their place. As these oxygen vacancies cluster together, they align and form conductive filaments between the top and bottom electrodes, switching the RRAM device from HRS to LRS. Conversely, the RESET process occurs when the polarity of the applied bias voltage is changed, leading to the breaking of conductive filaments in the HRS state. Some partially occupied oxygen vacancies below the top electrode are refilled with oxygen anions, converting them back into regular oxygen sites. Figure 7 provides a comparison of the switched resistance properties for Zn_1−x_Cu_x_O thin films in RRAM devices at different Cu concentrations. The RRAM device with 6% exhibits the highest ratio of LRS/HRS and a higher value of resistance in a state of HRS. When an external positive voltage is applied to the top electrode, the oxygen vacancies gather and arrange to connect the top and bottom metal electrodes, and electrons can be transferred near the oxygen vacancies by hopping, thus forming a conductive filament. On the contrary, when the polarity of the applied bias voltage is changed, some partially occupied oxygen vacancies under the top electrode are refilled by oxygen anions, resulting in the destruction of the oxygen vacancies forming the conductive filament. At this time, the device will change from a high current state to a low current state. The higher the number of destroyed oxygen vacancies, the higher the LRS/HRS ratio will be. Therefore, the highest LRS/HRS ratio of the 6% RRAM device should be attributed to the formation of conductive filaments and the destruction of more oxygen vacancies in 6% Cu-doped ZnO. The higher the number of oxygen vacancies destroyed, the lower the low current state, and, thus, the higher the LRS/HRS ratio. In addition, the grain size of Zn_1−x_Cu_x_O decreases with increasing Cu doping in the SEM analysis, which means there are more grain boundaries within the material. The grain boundaries typically exhibit higher electrical conductivity than the interior of grains in polycrystalline materials. The doping of copper can promote the formation of carrier transmission paths at the grain boundaries of Zn_1−x_Cu_x_O and can also increase the carrier concentration within the Zn_1−x_Cu_x_O material. This means that more carriers are available for transmission, which can lead to a lower resistive resistance state (LRS) and, thus, improve the LRS/HRS ratio and enhance the switching performance [29].

Figure 9 displays the lnI versus lnV plot of the Zn_1−x_Cu_x_O thin film, which is used to analyze the carrier conduction mechanism and understand the initial conductive filament formation process in the set/reset states of a metal–insulator–metal (MIM) structure. In the set state, the conduction path of the filaments is primarily influenced by the presence of oxygen vacancies in the Cu-doped ZnO films. The IV curves for the low resistance state (LRS) of all samples show slopes of approximately 0.98–0.99, indicating an ohmic conduction mechanism where the current is linearly dependent on the applied voltage. As the voltage increases, a stepwise increase in current is observed. On the other hand, in Figure 9b–d, the curves’ slopes in the high electric field region of the high resistance state (HRS) are approximately 7.27–8.2, and the current rises sharply with the voltage. This suggests that the accumulation of oxygen vacancies in Cu-doped ZnO films enables carriers to be transported through traps located at different energy levels, leading to the trap-assisted tunneling (TAT) mechanism responsible for this IV behavior [30,31,32]. Current density dominated by TAT can be expressed as:(1)$$J_{TAT}=AE^{2}exp\left[-\frac{B}{E}\right]$$
(2)$$A=\frac{q^{3}m}{16\pi^{2}\hbar m^{*}\phi t}$$
(3)$$B=\frac{4\sqrt{2m^{*}}}{3\hbar q}\phi_{t}^{3/2}$$
where $m^{*}$ is the effective mass of the electron, ℏ is Planck’s constant, *m* is the mass of the electron in free space, and ϕ*_t_* is the barrier height of the trap.

From the above results, the switching retention and endurance characteristics of the 6% Cu-doped ZnO thin film RRAM device are shown in Figure 10. It can be seen from the results that the 6% Cu-doped ZnO thin film RRAM device exhibits a stable state in more than one million switching cycles in Figure 10a. And the relationship between the resistance and switching time between the HRS and LRS states of the RRAM device can be clearly observed. RRAM devices also exhibit typical bipolar switching behavior and memory ratios greater than 10^3^ HRS/LRS. In addition, the endurance characteristics of the 6% Cu-doped ZnO thin film RRAM device are shown in Figure 10b. There is no significant change in the on/off ratio switching behavior cycling versus time curves of the RRAM device over 10^3^ s.

## 4. Conclusions

Zn_1−x_Cu_x_O films were deposited on FTO substrates fabricated using the sol-gel process. The crystal structure, optical and resistive switching properties of Zn_1−x_Cu_x_O were studied and analyzed. The intensity of the (002) peaks of the Zn_1−x_Cu_x_O increased with an increase in doping of Cu from x=0 to x=0.06. Moreover, as the Cu content increased, the intensity of the ultraviolet emission and the yellow–green light emission peaks increased in the photoluminescence spectrum of Zn_1−x_Cu_x_O film. It was found that the Zn_1−x_Cu_x_O thin film RRAM exhibits the ohmic conduction mechanism and the trap-assisted tunneling (TAT) mechanism in the low electric field region (LRS) and high electric field region (HRS), respectively. RRAM devices based on 6% Cu-doped ZnO films exhibited the highest LRS/HRS ratio and higher HRS state resistance.

## Figures and Tables

**Figure 1 nanomaterials-13-02685-f001:**
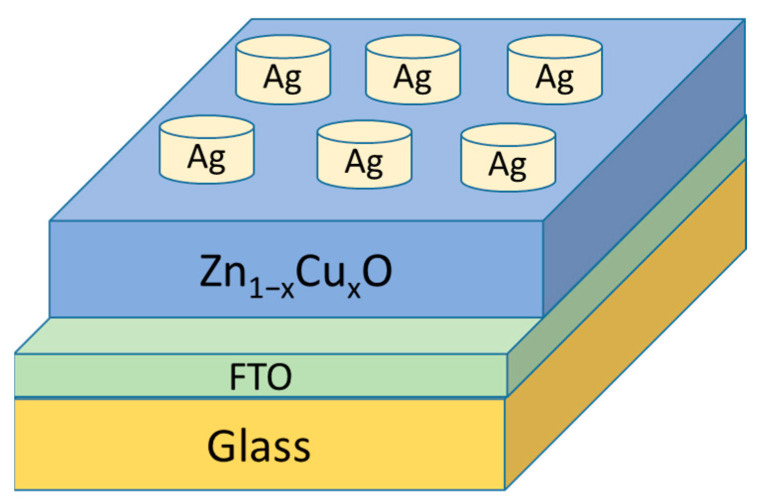
RRAM structure of Ag/Zn_1−x_Cu_x_O/FTO.

**Figure 2 nanomaterials-13-02685-f002:**
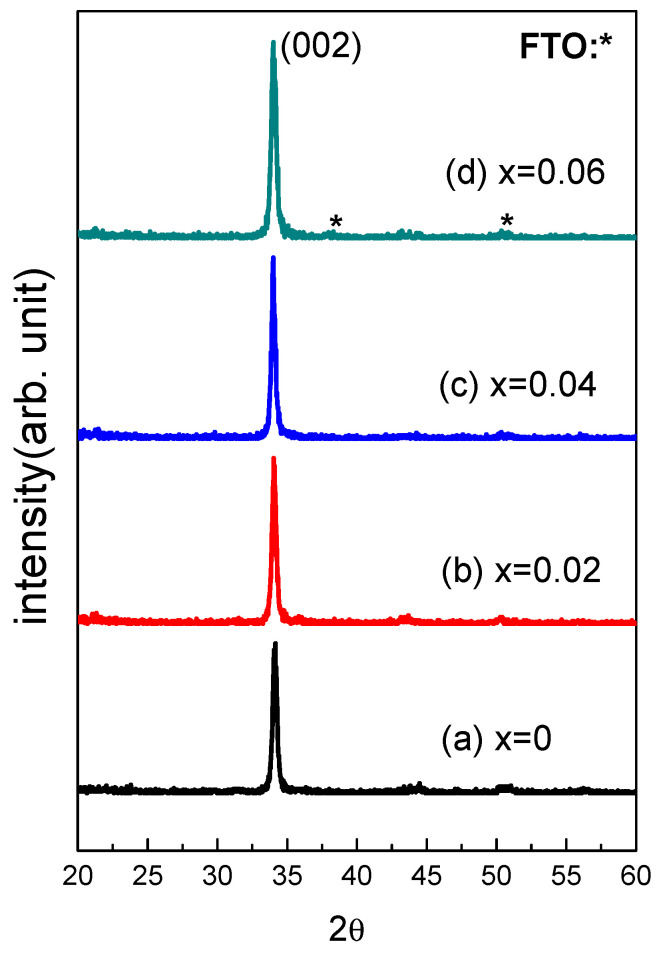
The patterns of XRD for Zn_1−x_Cu_x_O films.

**Figure 3 nanomaterials-13-02685-f003:**
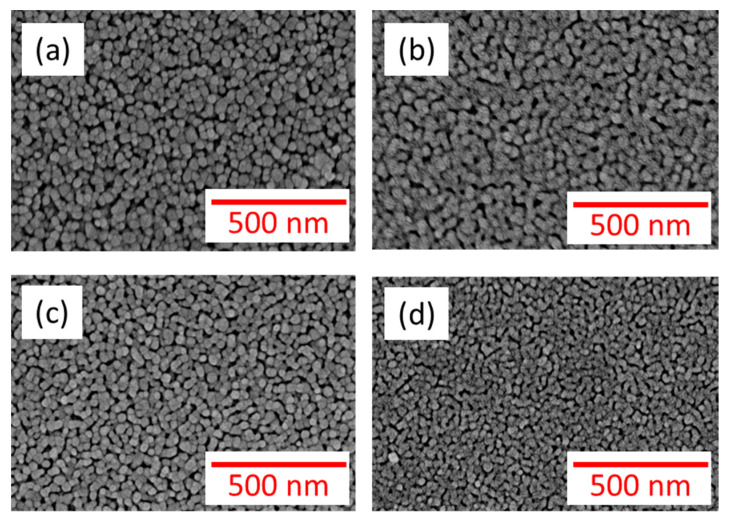
The images of SEM for Zn_1−x_Cu_x_O films of (**a**) x = 0, (**b**) x = 2%, (**c**) x = 4%, and (**d**) x = 6%.

**Figure 4 nanomaterials-13-02685-f004:**
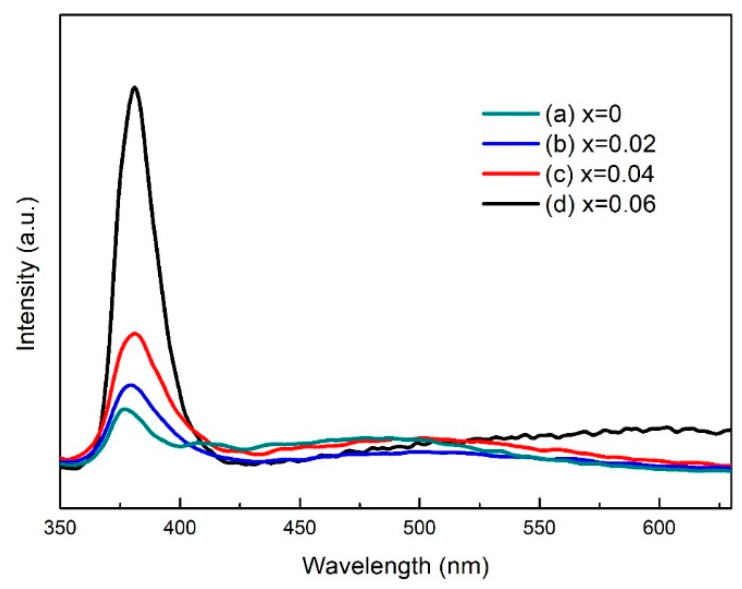
Photoluminescence (PL) spectra of Zn_1−x_Cu_x_O films of (a) x = 0, (b) x = 2%, (c) x = 4%, and (d) x = 6%.

**Figure 5 nanomaterials-13-02685-f005:**
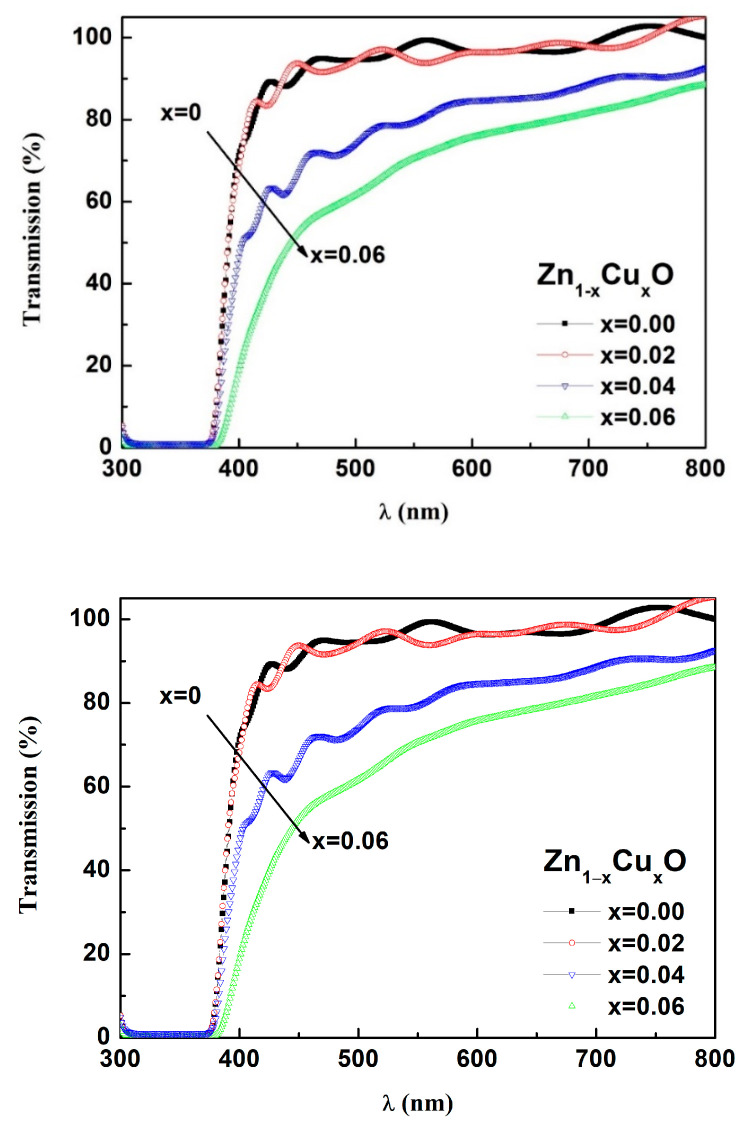
Spectra of optical transmittance for Zn_1−x_Cu_x_O.

**Figure 6 nanomaterials-13-02685-f006:**
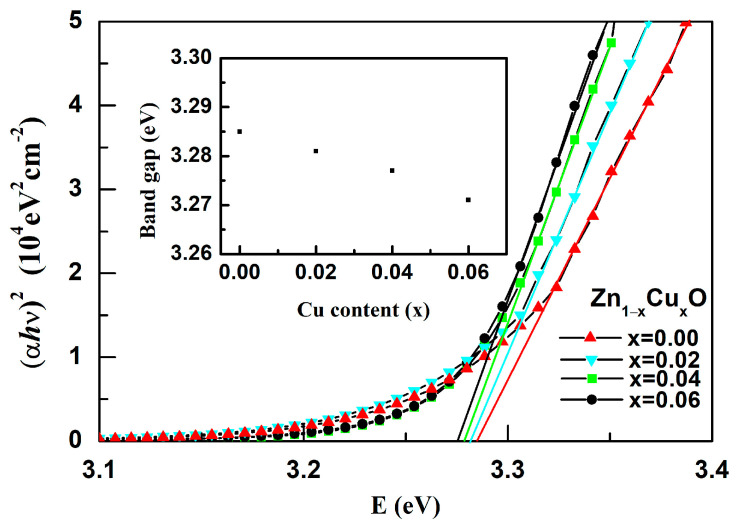
The curves of (*αhυ*)^2^ vs. Eg for Zn_1−x_Cu_x_O. The inset is Cu doping amount vs. bandgap energy.

**Figure 7 nanomaterials-13-02685-f007:**
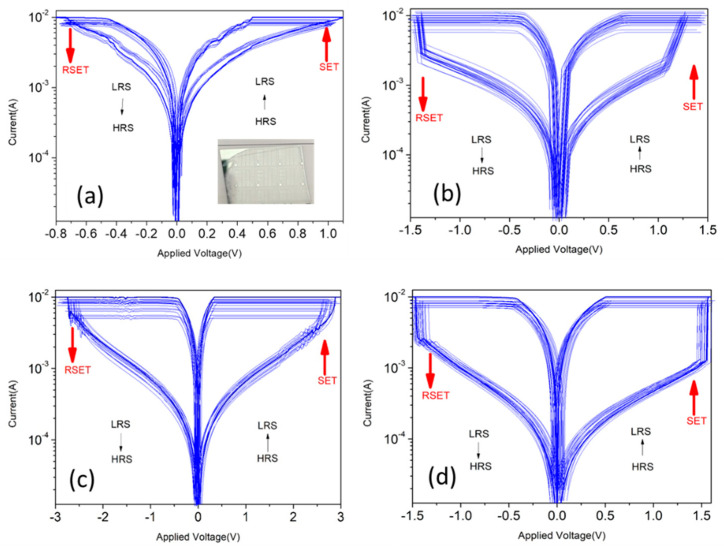
The switching curves of I–V of Zn_1−x_Cu_x_O RRAM of (**a**) x = 0, (**b**) x = 2%, (**c**) x = 4%, and (**d**) x = 6% (inset shows the real device image in (**a**)).

**Figure 8 nanomaterials-13-02685-f008:**
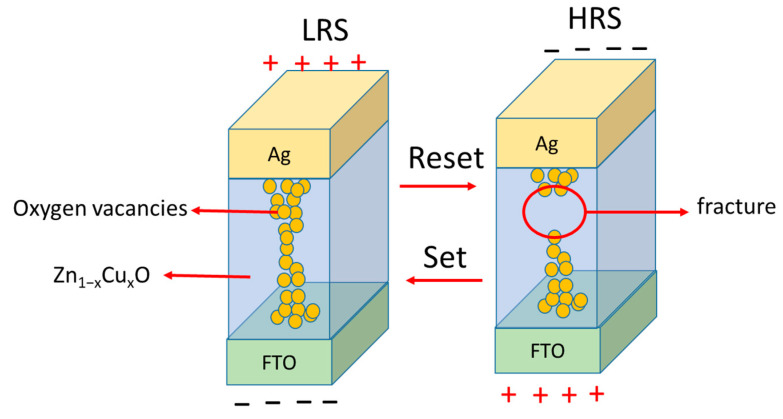
Schematic diagram of mechanisms of LRS and HRS for Ag/Zn_1−x_Cu_x_O/FTO RRAM device.

**Figure 9 nanomaterials-13-02685-f009:**
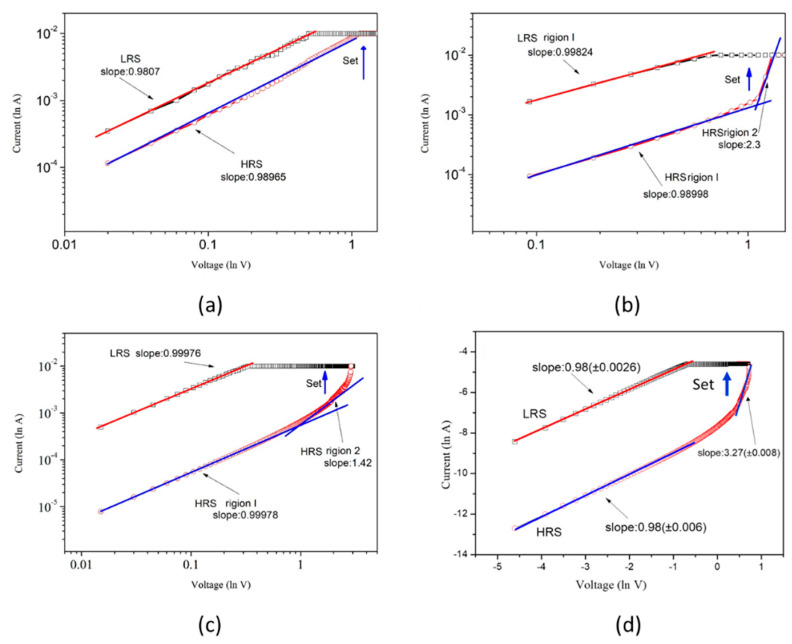
The curves of *I–V* for Zn_1−x_Cu_x_O RRAM for the *lnI–lnV* curve of (**a**) x = 0, (**b**) x = 2%, (**c**) x = 4%, and (**d**) x = 6%.

**Figure 10 nanomaterials-13-02685-f010:**
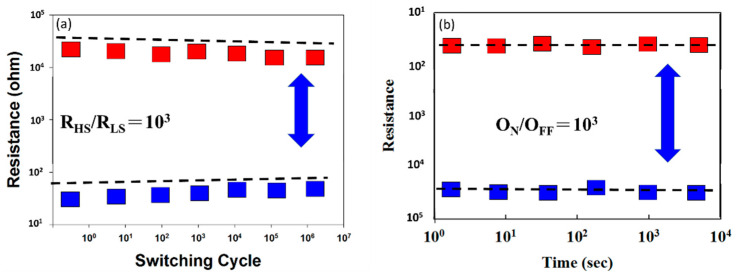
Switching endurance results of 6% Cu-doped ZnO film RRAM device showing (**a**) retention properties and (**b**) endurance properties.

## Data Availability

Not applicable.

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
