# Peer review of "Influences of Cu Doping on the Microstructure, Optical and Resistance Switching Properties of Zinc OxideThin Films"

_nanomaterials, 2023, doi:10.3390/nano13192685_

Round 1
Reviewer 1 Report
The manuscript reports the influence of Cu doping on properties of ZnO thin films, especially for resistance switching memory applications. The results shows some interesting properties of Cu-doped ZnO films, however, some discussions cannot be supported by experimental results and conclusion is not consistent. Based on the overall contents of the manuscript. I cannot recommend publication.
Here after, I summarize my specific comments on the manuscript.
1) Page 1 in abstract, the authors describe that the reduction of oxygen vacancies in the Zn1-xCuxO film contributes the improvement of RRAM devices. However, the oxygen vacancies increased with increasing the Cu doping concentration, for instance, as mentioned at page 4 line 111 based on the PL results. These are controversial.
2) Page 2, line 60. Please include explanation what is effects of FTO for the transmittance and how corrected the results.
3) Page 2, line 78. I cannot understand why the lattice strain introduced by Cu doping enhance the intensity of 002 diffraction peak. In addition, the peak intensity of 002 in Fig. 2 does not increase so much with Cu doping, and is nearly comparable.
3) Page 3, line 90, Authors explain the decrease of grain size with the increase of Cu concentration due to a smaller ionic radius of Cu compared to that of Zn. The discussion is not acceptable.
4) Page 4. The oxygen vacancies increased with increasing the Cu concentration based on the assignment of PL bands. This is controversial to the description in abstract.
5) Page 4, line 107. What is the interaction between the additional energy bands with the conduction or valence bands, resulting in a red-shift of the UV emission peak. Please discuss physics of the interaction in more details. The interaction decreases the bandgap?
6) On the other hand, the decrease of the bandgap is also explained by the smaller grain size of Cu doping. It is not clear why the band gap decreases due to the grain size. Together with the comments 5, the discussion regarding to the band gap depending on the Cu concentration is unclear.
7) Page 5, line 122.The meaning of “A” is not clear. In the manuscript, “the A is the effective mass and index of refractive function of holes and electrons”. Please correct the definition appropriately.
8) Regarding to the resistance switching property of ZnCuO in Fig 7, the set and reset voltage is strongly depending on the samples. Please explain the phenomena.
9) The highest ratio of LRS/HRS of the ZnO-based RRAM like as La-doped ZnO has been explained by the reduction of oxygen vacancies as mentioned in the introduction of the manuscript at page page 1 line 35. The authors also explain the improved LRS/HRS ration in ZnCuO by a reduction of oxygen vacancies. However, the authors report the increase of oxygen vacancies as increasing Cu doping concentration as mentioned above, and the transport mechanism of Cu-doped ZnO the trap-assisted tunneling owing to the increase of oxygen vacancies. The explanations are not consistent each other.
10) Something wrong? “enhanced photocatalytic activity in X-ray photoelectron spectroscopy” page 1 line 39.
Author Response
Dear Reviewer,
We appreciate your time and effort in reviewing our manuscript titled "Influences of Cu doping on the microstructure, optical, and resistance switching properties of Zinc Oxide thin films." We value your comments and feedback, which are instrumental in improving the quality of our work. We have carefully considered your comments and have made the necessary revisions to address your concerns. Here, we provide a detailed response to your specific comments:
1) Page 1 in abstract, the authors describe that the reduction of oxygen vacancies in the Zn1-xCuxO film contributes the improvement of RRAM devices. However, the oxygen vacancies increased with increasing the Cu doping concentration, for instance, as mentioned at page 4 line 111 based on the PL results. These are controversial.
Ans:
Thanks for reviewer's suggestions. The improvement of the 6% RRAM device should be attributed to the destruction of more oxygen vacancies in 6% Cu-doped ZnO, rather than the reduction of oxygen vacancies. It is due to that when an external positive voltage is applied to the top electrode, the oxygen vacancies gather and arrange to connect the upper and lower metal electrodes, and electrons can be transferred near the oxygen vacancies by hopping, thus forming a conductive filament. On the contrary, when the polarity of the applied bias voltage is changed, some partially occupied oxygen vacancies under the top electrode are refilled by oxygen anions, resulting in the destruction of the oxygen vacancies forming the conductive filament. At this time, the device will change from a high current state to a low current state. The higher the number of destroyed oxygen vacancies, the higher the LRS/HRS ratio will be. Therefore, the highest LRS/HRS ratio of the 6% RRAM device should be attributed to the destruction of more oxygen vacancies in 6% Cu-doped ZnO, rather than the reduction of oxygen vacancies. We are sorry that our mistake caused the reviewers to misunderstand. We have revised the explanation that the improved resistive switching performance of RRAM devices can be attributed to the formation of conductive filaments and the destruction of more oxygen vacancies in the Zn1-xCuxO film on page 1 line 20-21, page 7 line 197-208 and page 8 lines 225. In addition, we deleted the explanation that the accumulation of oxygen vacancies in the ZnO film contributes to the resistive switching performance in the PL analysis on page 4, line 111 of the original manuscript.
2) Page 2, line 60. Please include explanation what is effects of FTO for the transmittance and how corrected the results.
Ans:
Thanks for reviewer's suggestions. The effect of the fluorine-doped tin oxide (FTO) substrate was also taken into account during the transmittance correction. It is due to the FTO substrate can absorb light in the UV-visible range. It can lead to inaccurate transmittance measurements. We used a blank FTO substrate (without Zn1-xCuxO sample) for reference measurements. To correct for substrate effects, the transmittance of the FTO substrate alone was subtracted from the transmittance of the sample and substrate combination. This corrected transmittance data represents the transmittance of the Zn1-xCuxO sample without FTO substrate contribution. We have added explanations about the above in Page 2 line 66-71.
3) Page 2, line 78. I cannot understand why the lattice strain introduced by Cu doping enhance the intensity of 002 diffraction peak. In addition, the peak intensity of 002 in Fig. 2 does not increase so much with Cu doping, and is nearly comparable.
Ans:
Thanks for reviewer's suggestions. The increased intensity of the (002) diffraction peak at approximately 2θ = 34.5° is indicative of a more ordered and periodic arrangement of atoms in the crystal lattice along the c-axis. The lattice strain caused by the size mismatch between Cu and Zn ions induces this increased ordering. When the atomic planes along the c-axis are more regularly spaced, the X-rays are more effectively diffracted, resulting in a higher peak intensity in the XRD pattern. We have revised the description to make the explanation clearer on page 3 lines 92-97.
The (002) peak intensity of Zn1-xCuxO films increases with the increase of Cu doping content from 0% to 6%. (002) peak intensities are 240, 218, 200, and 180 for 0%, 2%, 4%, and 6%, respectively. The results showed that the peak intensity of (002) increased significantly.
4) Page 3, line 90, Authors explain the decrease of grain size with the increase of Cu concentration due to a smaller ionic radius of Cu compared to that of Zn. The discussion is not acceptable.
Ans:
Thanks for reviewer's suggestions. The grain size in Zn1-xCuxO films decreases with increasing Cu concentration (from 0% to 6%), which can be attributed to the crystal structure and growth of the material. When copper is introduced into the zinc oxide lattice, (Cu) and zinc (Zn) have different atomic sizes and lattice constants, creating a lattice mismatch. The lattice mismatch in the Zn1-xCuxO film causes crystal strain in the crystal structure and hinders the growth of larger grains. We have revised the description to make the explanation clearer on page 3 lines 106-110.
4) Page 4. The oxygen vacancies increased with increasing the Cu concentration based on the assignment of PL bands. This is controversial to the description in abstract.
Ans:
Thanks for reviewer's suggestions. We have revised the explanation in line 20 of page 1 of abstract that the improved resistive switching performance of RRAM devices can be attributed to the formation of conductive filaments and the destruction of more oxygen vacancies in the Zn1-xCuxO film. In addition, we deleted the explanation that the accumulation of oxygen vacancies in the ZnO film contributes to the resistive switching performance in the PL analysis on page 4, line 111 of the original manuscript.
5) Page 4, line 107. What is the interaction between the additional energy bands with the conduction or valence bands, resulting in a red-shift of the UV emission peak. Please discuss physics of the interaction in more details. The interaction decreases the bandgap?
Ans:
Thanks for reviewer's suggestions. When copper ions (Cu) are replaced into the ZnO lattice to form Zn1-xCuxO, defect states are created within the band gap, effectively introducing additional energy levels. The additional energy levels and energy differences between energy bands effectively change the energy required for electron transitions. The emitted photons therefore have lower energy, corresponding to longer wavelengths in the visible spectrum (red shift). In addition, due to the existence of defect states, the effective band gap that electrons need to overcome becomes smaller. This means that less energy required to lift electrons from the valence band to the conduction band through these defect states is equivalent to a band reduction. We have added the description about the additional energy bands on page 4 lines 126-135.
6) On the other hand, the decrease of the bandgap is also explained by the smaller grain size of Cu doping. It is not clear why the band gap decreases due to the grain size. Together with the comments 5, the discussion regarding to the band gap depending on the Cu concentration is unclear.
Ans:
Thanks for reviewer's suggestions. It is due to when copper ions (Cu) are replaced within the band gap in the ZnO lattice, defects are created and additional energy levels are introduced. The energy difference between the additional energy levels and the energy bands changes the energy required for the electron to transition, making the effective band gap that the electron needs to overcome smaller. Furthermore, the spatial confinement of carriers increases with decreasing grain size. Electrons in these confined states may require less energy to transition between energy levels within the grain, effectively reducing the effective band gap for charge carriers within the grain. We have added the description on page 5 lines 155-162.
7) Page 5, line 122. The meaning of “A” is not clear. In the manuscript, “the A is the effective mass and index of refractive function of holes and electrons”. Please correct the definition appropriately.
Ans:
Thanks for reviewer's suggestions. We have corrected the definition about the meaning of “A” in page 5 line 147-148.
8) Regarding to the resistance switching property of ZnCuO in Fig 7, the set and reset voltage is strongly depending on the samples. Please explain the phenomena.
Ans:
Thanks for reviewer's suggestions. According to the bipolar I-V switching curve of the Zn1-xCuxO film in Figure 7, the set voltages are 1.1V (0%), 1.2V (2%), 3V (4%) and 1.5V (6%) respectively. In addition, the rest voltages are -0.9V(0%), -1.4V(2%), -2.8V(4%) and -1.5V(6%) respectively. We have added the description on page 6 lines 180-182. But the differences in set and reset voltages observed in Zn1-xCuxO films with different Cu contents may be due to complex interactions between the dopant (Cu), material properties, and device properties. The specific mechanisms involved require future detailed analysis of material properties and device behavior.
9) The highest ratio of LRS/HRS of the ZnO-based RRAM like as La-doped ZnO has been explained by the reduction of oxygen vacancies as mentioned in the introduction of the manuscript at page page 1 line 35. The authors also explain the improved LRS/HRS ration in ZnCuO by a reduction of oxygen vacancies. However, the authors report the increase of oxygen vacancies as increasing Cu doping concentration as mentioned above, and the transport mechanism of Cu-doped ZnO the trap-assisted tunneling owing to the increase of oxygen vacancies. The explanations are not consistent each other.
Ans:
When an external positive voltage is applied to the top electrode, the oxygen vacancies gather and arrange to connect the upper and lower metal electrodes, and electrons can be transferred near the oxygen vacancies by hopping, thus forming a conductive filament. On the contrary, when the polarity of the applied bias voltage is changed, some partially occupied oxygen vacancies under the top electrode are refilled by oxygen anions, resulting in the destruction of the oxygen vacancies forming the conductive filament. At this time, the device will change from a high current state to a low current state. The higher the number of destroyed oxygen vacancies, the higher the LRS/HRS ratio will be. Therefore, the highest LRS/HRS ratio of the 6% RRAM device should be attributed to the destruction of more oxygen vacancies in 6% Cu-doped ZnO, rather than the reduction of oxygen vacancies.
We have revised the explanation that the improved resistive switching performance of RRAM devices can be attributed to the formation of conductive filaments and the destruction of more oxygen vacancies in the Zn1-xCuxO film on page 1 line 20-21, page 7 line 197-208 and page 8 lines 225. In addition, we deleted the explanation that the accumulation of oxygen vacancies in the ZnO film contributes to the resistive switching performance in the PL analysis on page 4, line 111 of the original manuscript.
10) Something wrong? “enhanced photocatalytic activity in X-ray photoelectron spectroscopy” page 1 line 39.
Ans:
Thanks for reviewer’s comments, we have corrected the description in page 1 line 38-39.
Your feedback has been invaluable in guiding these improvements, and we sincerely thank you for your dedication to ensuring the quality of our work. Your expert assessment will be greatly appreciated. Once again, thank you for your constructive feedback.

Reviewer 2 Report
My comments are as follows:
This paper shows the influences of Cu doping on several properties including the optical, and resistance switching properties of ZnO thin films. This experimental result is so interesting covering from materials synthesis to electronic devices. However, the presented data cannot fully support the claim. This work can be accepted after addressing the following issues.
(1) It is more necessary to shows the real optical image in this experiment.
(2) It is more necessary to shows the endurance and retention data of Zn1-xCuxO RRAM device.

Author Response
Dear Reviewer,
We appreciate your time and effort in reviewing our manuscript titled "Influences of Cu doping on the microstructure, optical, and resistance switching properties of Zinc Oxide thin films." We value your comments and feedback, which are instrumental in improving the quality of our work. We have carefully considered your comments and have made the necessary revisions to address your concerns. Here, we provide a detailed response to your specific comments:
(1)It is more necessary to show the real optical image in this experiment.
Ans: Thanks for the reviewer's suggestions. We have added the real optical image of the RRAM device in Figure 7 (a).
(2) It is more necessary to shows the endurance and retention data of Zn1-xCuxO RRAM device.
Ans: Thanks for the reviewer's suggestions. We have added the endurance and retention data and explanations for RRAM device in Figure 10 and on page 9, lines 238-246.
Your feedback has been invaluable in guiding these improvements, and we sincerely thank you for your dedication to ensuring the quality of our work. Your expert assessment will be greatly appreciated. Once again, thank you for your constructive feedback.

Reviewer 3 Report
The article corresponds to the subject of the journal. It presents the results of studying the memristive properties of structures based on ZnO films doped with Cu. Such materials are of considerable interest for the creation of microelectronic devices.
The article is well structured and contains a number of new and interesting results on the effect of copper doping on the structural (grain size), optical (photoluminescence intensity) and electrophysical properties of the obtained films.
At the same time, the presented text does not fully describe the experimental technique that is important for understanding the presented results:
1. In the experimental part, it is necessary to describe in more detail the procedure for preparing and depositing films: what solution and what equipment were used to deposit the films, how and how many Ag contacts were deposited on the film, and what is the geometry of the device used in electrophysical experiments.
2. It is necessary to provide data on the stability of memristive properties in the obtained devices. How many LRS-HRS switching cycles are possible in them without losing their electrophysical properties. Without these data, it is not possible to evaluate the possibility of using the obtained materials in RRAM devices.
3. It is desirable to depict the results for all four samples in the same way in Figure 7. It is necessary to add in the caption to this figure information about which samples correspond to figures (a), (b), (c), (d).
After answering the questions posed and making appropriate clarifications, the article requires a second review.
Author Response
Dear Reviewer,
We appreciate your time and effort in reviewing our manuscript titled "Influences of Cu doping on the microstructure, optical, and resistance switching properties of Zinc Oxide thin films." We value your comments and feedback, which are instrumental in improving the quality of our work. We have carefully considered your comments and have made the necessary revisions to address your concerns. Here, we provide a detailed response to your specific comments:
1.In the experimental part, it is necessary to describe in more detail the procedure for preparing and depositing films: what solution and what equipment were used to deposit the films, how and how many Ag contacts were deposited on the film, and what is the geometry of the device used in electrophysical experiments.
Ans:
Thanks for reviewer's suggestions. We have added a more detailed description about the process of preparing and depositing thin films on page 2 lines 50-61. One hundred circular top silver electrodes (100 nm thickness and 0.05 cm diameter) were fabricated on top of the Zn1-xCuxO layer using thermal evaporation technique at 5 × 10-6 torr. We also added the description of the Ag contacts and device geometry on page 2, lines 76-78.
- It is necessary to provide data on the stability of memristive properties in the obtained devices. How many LRS-HRS switching cycles are possible in them without losing their electrophysical properties. Without these data, it is not possible to evaluate the possibility of using the obtained materials in RRAM devices.
Ans:
Thanks for reviewer's suggestions. We have provided data on the stability of the memristive properties for the obtained devices in Figure 7(a)~(d). RRAM devices can be operated for LRS-HRS switching cycles up to 100 times without losing their electrophysical properties. We also added the description on page 6, lines 174-176.
- It is desirable to depict the results for all four samples in the same way in Figure 7. It is necessary to add in the caption to this figure information about which samples correspond to figures (a), (b), (c), (d).
Ans: Thanks for reviewer's suggestions. We have added information about which samples correspond to curves of (a), (b), (c), (d) in Figure 7.
Your feedback has been invaluable in guiding these improvements, and we sincerely thank you for your dedication to ensuring the quality of our work. Your expert assessment will be greatly appreciated. Once again, thank you for your constructive feedback.

Round 2
Reviewer 3 Report
The authors made the necessary clarifications to the text of the article. It may be published in its presented form.
Author Response
Dear Reviewer,
We would like to express our sincere gratitude for your thoughtful review of our manuscript. We are pleased to hear that you found the necessary clarifications to the manuscript satisfactory and that you recommend the publication of our work in its presented form. Once again, we would like to extend our appreciation for your time and effort in reviewing our work.
Thank you for your support and valuable input.
Sincerely,